# Multi-input CRISPR/Cas genetic circuits that interface host regulatory networks

Alec AK Nielsen & Christopher A Voigt[*]

## Abstract

Genetic circuits require many regulatory parts in order to implement signal processing or execute algorithms in cells. A potentially scalable approach is to use dCas9, which employs small guide RNAs (sgRNAs) to repress genetic loci via the programmability of RNA:DNA base pairing. To this end, we use dCas9 and designed sgRNAs to build transcriptional logic gates and connect them to perform computation in living cells. We constructed a set of NOT gates by designing five synthetic *Escherichia coli* $\sigma_{70}$ promoters that are repressed by corresponding sgRNAs, and these interactions do not exhibit crosstalk between each other. These sgRNAs exhibit high on-target repression (56- to 440-fold) and negligible off-target interactions (< 1.3-fold). These gates were connected to build larger circuits, including the Boolean-complete NOR gate and a 3-gate circuit consisting of four layered sgRNAs. The synthetic circuits were connected to the native *E. coli* regulatory network by designing output sgRNAs to target an *E. coli* transcription factor (*malT*). This converts the output of a synthetic circuit to a switch in cellular phenotype (sugar utilization, chemotaxis, phage resistance).

**Keywords** CRISPR; genetic compiler; synthetic biology; TALE; TetR homologue
**Subject Categories** Synthetic Biology & Biotechnology; Methods & Resources
**Mol Syst Biol.** (2014) 10: 763

## Introduction

Genome editing has been revolutionized by the RNA-guided endonuclease Cas9 from *Streptococcus pyogenes* due to its ability to target DNA sequences adjacent to "NGG" motifs using a guide RNA (Cong *et al*, 2013; Esvelt *et al*, 2013; Jiang *et al*, 2013; Shalem *et al*, 2013; Wang *et al*, 2013; Zhou *et al*, 2014). This programmability has been harnessed for gene regulation using a Cas9 double mutant that eliminates nuclease activity (dCas9) so that guide RNAs cause it to bind tightly to the corresponding DNA sequence without cleaving it (Jinek *et al*, 2012). This complex can serve as a repressor by blocking RNAP binding to a promoter or by terminating transcription (Bikard *et al*, 2013; Esvelt *et al*, 2013; Qi *et al*, 2013). A chimeric small guide RNA (sgRNA) is sufficient to drive Cas9 to a target (Jinek *et al*, 2012), and it comprises a complementary domain that binds to the DNA followed by a "handle" that is bound by Cas9. Considering the programmability of DNA:RNA interactions and the existence of a "seed" region at the 3′-end of the sgRNA's complementary region, this system could yield ~$10^7$ orthogonal sgRNA:DNA pairs. This is a potentially versatile platform for building genetic circuits, which have been limited in size and sophistication by the number of available orthogonal transcription factors.

Extensible circuits, whose inputs and outputs are of an identical form, can be connected in different ways in order to perform user-defined computational operations (Nielsen *et al* 2013). For genetic circuits, the simplest way to achieve this is to design gates with inputs and outputs that are both promoters (Tamsir *et al*, 2011; Moon *et al*, 2012; Stanton *et al*, 2014). In this formalism, the common signal carrier is RNAP flux and gates are connected by having the output of one serve as the input to the next. The majority of transcriptional gates have been built using DNA-binding proteins. The challenge has been to obtain large sets of orthogonal proteins that do not cross-react with each other's binding sites. These sets can be obtained either by part mining, where bioinformatics is applied to search databases for classes of regulators that are synthesized and screened (Moon *et al*, 2012; Rhodius *et al*, 2013; Stanton *et al*, 2014), or by building variants of modular DNA-binding proteins whose domains can be engineered to target different operators [e.g. ZFPs (Beerli & Barbas, 2002; Miller *et al*, 2007) and TALEs (Morbitzer *et al*, 2010; Miller *et al*, 2011)]. For both approaches, cross-reactions are prevalent and many variations have to be screened to obtain an orthogonal core set. Another challenge is that within a regulator class, some can be non-toxic whereas others exhibit extreme toxicity (Kimelman *et al*, 2012; Stanton *et al*, 2014). Collectively, restrictions on function, orthogonality, and toxicity reduce the size of the libraries dramatically; for example, an initial set of 73 TetR homologues was reduced to 16 repressors (Stanton *et al*, 2014).

Here, we present a set of transcriptional gates based on sgRNA-guided repression of synthetic *Escherichia coli* $\sigma_{70}$ promoters (Fig 1A). The input to an sgRNA NOT gate is a promoter that contains a precise transcription start site (+1) so that additional nucleotides are not added to the 5′-end of the sgRNA, which has

---

Synthetic Biology Center, Department of Biological Engineering, Massachusetts Institute of Technology, Cambridge, MA, USA
  *Corresponding author. Tel: +1 617 324 4851; E-mail: cavoigt@gmail.com

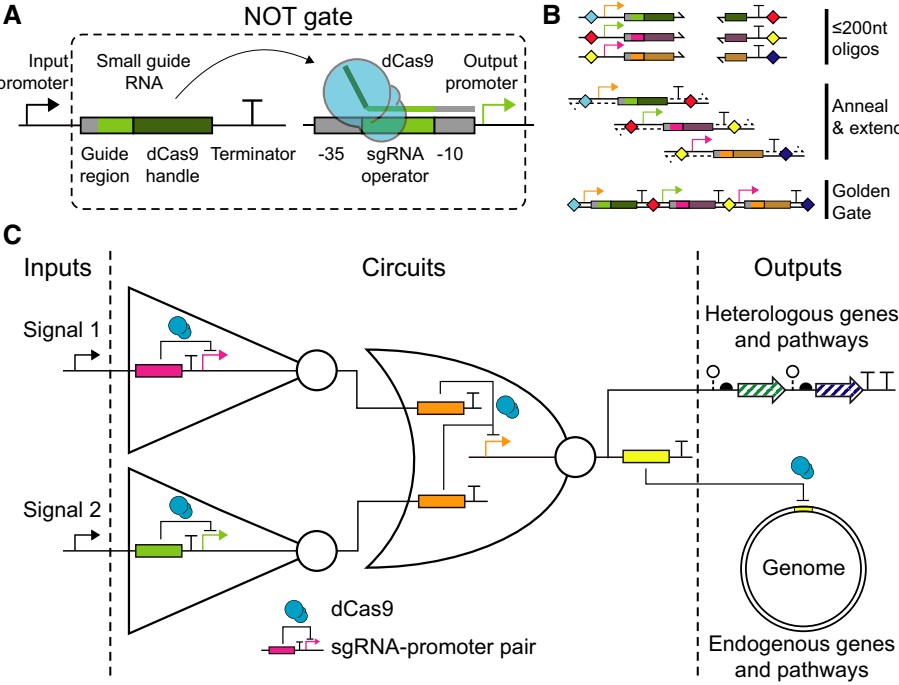

**Figure 1. Schematics of dCas9 logic circuit design and construction.**

A   CRISPR/Cas-based NOT gates comprise a catalytically dead dCas9 protein, an input promoter that transcribes a small guide RNA (sgRNA), and a synthetic output promoter with an sgRNA operator between the −35 and −10 sigma factor binding sites. When the dCas9 handle of the sgRNA (dark green) complexes with dCas9 (blue), the sgRNA binds the operator (light green) and a sigma factor binding site (gray), causing steric repression of transcription initiation at the output promoter.

B   CRISPR/Cas genetic circuits are easily constructed from pairs of ssDNA oligonucleotides ≤ 200 nt long that encode the necessary genetic parts (promoter, sgRNA, terminator, assembly scars, and restriction enzyme recognition sites). These oligos are annealed to each other at the dCas9 handle and extended. The resulting dsDNA modules are assembled in a one-pot Golden Gate assembly reaction. (Colored diamonds are assembly scars.)

C   Genetic circuits that respond to chemical input signals can be constructed from simple NOT and NOR gate motifs. In these circuits, dCas9 (blue) mediates repression of synthetic promoters by programmable sgRNAs (visualized as solid colored rectangles from here on). Both heterologous and endogenous genes can be regulated at circuit outputs by expressing sgRNAs tailored to target transcription initiation or elongation.

been shown to reduce activity (Larson *et al*, 2013). The sgRNA includes a guide region that targets dCas9 to the cognate bacterial promoter. A strong terminator (Chen *et al*, 2013; Qi *et al*, 2013) is placed after the sgRNA to stop transcription. The output of the gate is an *E. coli* constitutive promoter (BBa_J23101) that has been modified to include both forward and reverse "NGG" PAMs (for targeting either the template or non-template strands of the promoter), and a unique 13 bp "operator" region between the −35 and −10 $\sigma_{70}$ binding sites (Fig 2C). The entire transcription unit (promoter, sgRNA, and terminator) can be constructed from a pair of ≤ 200 nt single-stranded DNA oligonucleotides that are annealed and extended at the dCas9 handle region. These ssDNA oligos also encode Type IIs restriction enzyme recognition sites that flank the transcription unit. The resulting dsDNA modules can then be combined into a final circuit plasmid using a one-pot Golden Gate assembly reaction (Engler *et al*, 2009) (Fig 1B).

Multi-input NOR and NAND gates are "Boolean-complete" and are each sufficient to build any user-defined digital computational operation (Katz & Boriello, 2004). Transcription factor-based NOR gates have previously been built by placing two input promoters in series upstream from a repressor gene (Tamsir *et al*, 2011; Stanton *et al*, 2014). Without additional RNA processing, this design does not work for sgRNA circuits because of the detrimental influence of 5′-mismatches and the "roadblocking" effect of CRISPRi, which is

small for template-targeting sgRNAs and substantial for non-template-targeting sgRNAs (Qi *et al*, 2013). Hammerhead ribozymes and endoRNase cleavage of 5′-mismatches have both been shown to effectively remove extraneous 5′-RNA from sgRNAs (Gao & Zhao, 2014; Nissim *et al*, 2014) and could be employed in multi-input dCas9 circuits. Instead, our design is based on two transcription units per NOR gate, each of which contains a different input promoter. When either promoter is active, the sgRNA is transcribed and represses the output promoter. This design allows larger circuits to be constructed simply by changing the pattern of input and output promoters around the sgRNAs. This approach requires that the sgRNAs be able to be layered into a cascade, which has been shown to work in mammalian cells (Kiani *et al*, 2014; Nissim *et al*, 2014).

Linking the output(s) of a genetic circuit to regulate host genes provides control over cellular responses. For example, cells could be programmed to sense the cell density in a fermenter and respond by expressing enzymes to redirect flux through global metabolism (Nielsen *et al*, 2014). Similarly, the cell phenotype could be controlled, like the ability to swim or associate into biofilms. Various approaches have been taken to link synthetic circuits to endogenous genes. For example, MAGE has been used to insert T7 RNAP promoters upstream from genes participating in lycopene biosynthesis in order to upregulate production by expressing the polymerase

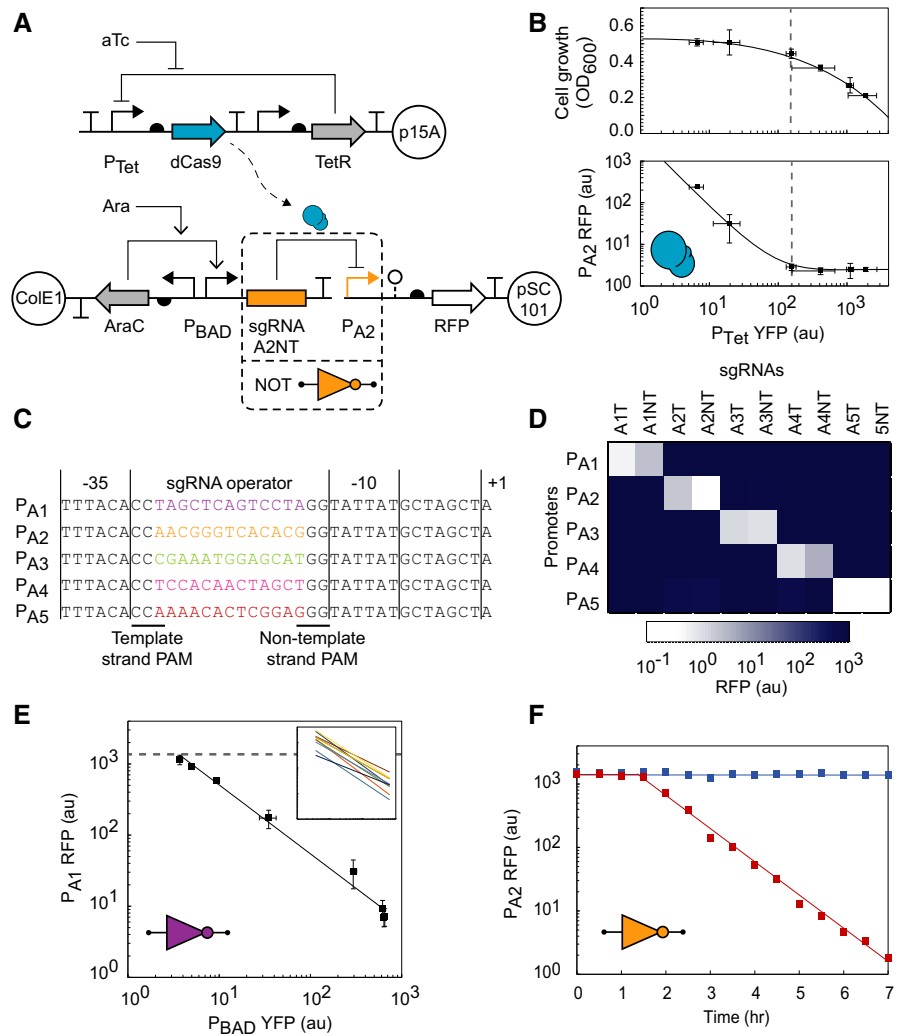

**Figure 2.    Characterization of dCas9 and orthogonal sgRNA NOT gates.**

A    The inducible dCas9 and sgRNA system comprises a medium-copy plasmid with $P_{Tet}$-inducible dCas9, a high-copy plasmid with $P_{BAD}$-inducible sgRNAs, and a low-copy plasmid encoding a synthetic sgRNA-repressible promoter driving RFP.

B    When sgRNA-A2NT is constitutively induced, increasing dCas9 expression causes greater repression of $P_{A2}$ (lower panel), at the cost of decreased cell growth (upper panel). All samples were grown in the presence of 2 mM arabinose. Concentrations of aTc used from left to right (ng/ml): 0.0391, 0.313, 0.625, 1.25, 5, and 10. A single intermediate expression value for dCas9 was used for the remaining experiments (0.625 ng/ml aTc, dashed lines).

C    Synthetic repressible promoters designed by modifying the sequence of promoter BBa_J23101. The −35 and −10 $\sigma_{70}$ binding sites flank forward and reverse "NGG" protospacer adjacent motifs (PAMs) and a promoter-specific 13 bp sgRNA operator. An sgRNA bound to dCas9 will base pair with one strand of a promoter's sgRNA operator and one of the $\sigma_{70}$ binding sites, causing steric repression of transcription initiation. In the absence of repression, transcription of the downstream RNA begins at the +1 site.

D    The crosstalk map for all combinations of sgRNAs and synthetic promoters is shown. The heat map indicates the amount of RFP observed for that sgRNA-promoter pair. Only cognate pairs of sgRNAs and promoters exhibit significant repression, whereas non-cognate pairs interact negligibly. Samples were grown in the presence of 0.625 ng/ml aTc and 2 mM arabinose.

E    The response function for sgRNA-A1T measured by expressing intermediate levels of sgRNA-A1T reveals a non-cooperative, log-linear relationship between the input and output promoters. The solid line visualizes a power law fit to the data points. Error bars represent the standard deviation of fluorescence geometric mean for three independent experiments on different days. The reporter expression when dCas9 is not induced is shown (dashed line), and all other samples were grown in the presence of 0.625 ng/ml aTc. Concentrations of arabinose used from left to right (mM): 0, 0.0313, 0.0625, 0.125, 0.25, 0.5, 1, and 2. Inset: The power law fits for each of the 10 sgRNAs and their cognate promoters (data presented in Supplementary Fig S3); axes values are the same as the encompassing figure.

F    The temporal dynamics of dCas9 and sgRNA induction are shown. Red squares indicate induction of both dCas9 (0.625 ng/ml aTc) and sgRNA-A2NT (2 mM arabinose) commencing at $t = 0$ h. Blue squares indicate uninduced cultures. After a ~90-min delay, fluorescence decreases concomitantly with cell dilution—occurring at a rate of 33 min per doubling.

as a circuit output (Wang *et al*, 2009). Natural and synthetic sRNAs have been used to knockdown endogenous genes involved in motility (Sharma *et al*, 2013), iron metabolism (Kang *et al*, 2012), acetone formation (Tummala *et al*, 2003), β-glucuronidase (Man *et al*, 2011), membrane porin and flagellin genes (Sharma *et al*, 2012), and to increase tyrosine and cadaverine production (Na *et al*,

2013). Finally, strains have been constructed that express a protein that can be targeted to the genome (ZFP: Beerli & Barbas, 2002; TALE: Morbitzer *et al*, 2010; Zhang *et al*, 2011; or dCas9: Farzadfard *et al*, 2013; Gilbert *et al*, 2013; Qi *et al*, 2013) to upregulate or knockdown endogenous genes. Here, we link synthetic dCas9-based circuits to the native *E. coli* regulatory network by designing the final sgRNA in a circuit to target a transcription factor on the host genome. This provides a generalizable mechanism by which the same biochemistry is used to both perform computation and also actuate host phenotype in response to conditions defined by the circuitry (Fig 1C).

## Results

### Orthogonal NOT gates based on dCas9 and sgRNAs

A three-plasmid system was built to measure sgRNA orthogonality and characterize their performance in the context of a NOT gate (Fig 2A). The first plasmid controls the expression of *S. pyogenes* dCas9 from an aTc-inducible $P_{Tet}$ promoter. The sgRNA is carried on a high-copy plasmid and transcribed using a variant of the arabinose-inducible $P_{BAD}$ promoter that is truncated to end at the transcription start site (+1). Finally, the output promoter repressed by the dCas9-sgRNA complex is transcriptionally fused to red fluorescent protein (RFP) and carried on a low-copy plasmid.

dCas9 can exhibit toxicity when overexpressed. To reduce background expression, we selected an aTc-inducible $P_{Tet}$ variant that exhibits low leakiness and added the strong L3S3P21 terminator (Chen *et al*, 2013) upstream to insulate from read-through transcription on the plasmid backbone. As the expression of dCas9 is increased, higher fold repression is observed, but this comes at the cost of reduced cell growth (Fig 2B). These effects are balanced at 0.625 ng/ml aTc, which elicits near-full repression with a growth impact of < 15% (after 6 h, an $OD_{600}$ of 0.44 versus 0.51). This induction level is used for all subsequent experiments.

A set of five synthetic promoters ($P_{A1}$–$P_{A5}$) were designed to be targeted by corresponding sgRNAs. An *E. coli* constitutive promoter (BBa_J23101) was chosen as a scaffold, and the operator that is recognized by the sgRNA was inserted between the −35 and −10 consensus sites where the housekeeping $\sigma_{70}$ binds (Fig 2C). The region between these sites is 17 bp, the center of which contains a unique 13 bp sequence that is bound by the "seed" of the sgRNA complementary region, which is less tolerant of RNA:DNA mismatches (Jinek *et al*, 2012). This is flanked by forward and reverse "NGG" protospacer adjacent motifs (PAMs), which are required for dCas9 binding (Marraffini & Sontheimer, 2010). When dCas9 is directed to this region by a corresponding sgRNA, the promoter is repressed by dCas9 sterically blocking the binding of *E. coli* RNAP. The orthogonal sgRNAs (sgRNA-A1–sgRNA-A5) were designed by selecting distinct 13 bp seed sequences that have no matches to PAM-proximal sequences in the *E. coli* genome. Two variants of each sgRNA were built that target the non-template (—NT) and template (—T) strands of each promoter. Each of the sgRNAs strongly represses its target promoter (56- to 440-fold), with no preference for the non-template or template strand, as observed previously (Bikard *et al*, 2013). The orthogonality of the promoters and sgRNAs are near perfect, with essentially no off-target interactions

(Fig 2D). In addition, we observe only a small amount of toxicity when the sgRNAs are highly expressed, and no growth differences between the sgRNA variants (Supplementary Figures S1, S2, S3 and S4 and Supplementary Tables S1 and S2).

The response function of a gate captures how the output changes as a function of input. This is critical in predicting how gates can be connected to form larger circuits. To characterize the gates, the $P_{BAD}$ promoter serves as the input, which we characterized separately as a function of arabinose concentration. This is used to rescale the data to report it as a function of promoter activity, as opposed to inducer concentration (Fig 2E). The log-linear shape of this response curve is approximated well by a power law and is very different from those observed from similar gates based on transcription factors, which saturate as a Langmuir isotherm. This log-linearity is also evident when observing the relationship between the intermediate and output promoters of an sgRNA cascade (Fig 3B, right).

The dynamics of repression were also measured (Fig 2F). After induction, there is an initial delay of 1.5 h corresponding to the activation of $P_{Tet}/P_{BAD}$ and the accumulation of dCas9/sgRNA. After this delay, there is a consistent exponential decline in RFP ($t_{1/2}$ = 33 min) over 7 h, which is consistent with the dilution rate of the reporter expected from cell division.

### Circuits based on layered sgRNA gates

The advantage of transcriptional gates is that they can be easily interconnected in order to build more complex circuit functions. Gates where repression is based on a non-coding RNA (ncRNA) can be challenging to connect in series for three reasons. First, they require more precision in the promoter start site or additional RNA processing due to sensitivities in the addition or removal of nucleotides at the 5′-end. Second, changing the ribosome binding site (RBS) has been an important lever for functionally connecting protein-based gates. The RBS is not relevant for an ncRNA-based gate, and matching gate responses by promoter tuning is more challenging. This is exacerbated by the shape of the response functions for the sgRNA-based gates, which do not plateau at high- or low-input promoter levels (Fig 2E); therefore, the input to any gate needs to have a very wide dynamic range in order to avoid signal degradation at each layer. However, despite these challenges, sgRNA-mediated repression has desirable properties that other ncRNA technologies do not possess, such as high dynamic range, specificity, and the ability to be composed into cascades (Qi & Arkin, 2014).

The layering of two NOT gates based on sgRNAs has been previously demonstrated in mammalian cells (Kiani *et al*, 2014; Nissim *et al*, 2014). We built a similar circuit architecture by connecting two of our sgRNA-based gates in series in *E. coli* (Fig 3A). These were connected simply by combining the parts from the sgRNA-A2NT and sgRNA-A4NT gates in the appropriate order with no additional tuning. dCas9 is induced from a low-leakage variant of $P_{Tet}$, as was done for the characterization of individual gates. In the absence of dCas9, the background activity of the output promoter ($P_{A4}$) is 1,040 au (arbitrary units, Fig 3B, leftmost bar). When dCas9 is induced, this resulted in a 98-fold repression of the circuit output ($P_{A4}$) compared to no sgRNA production (Fig 3B, left). When the circuit's input promoter is induced with DAPG, the output state recovers completely to the level of the dCas9 (—) control. By observing the middle promoter ($P_{A2}$) in the cascade in a separate

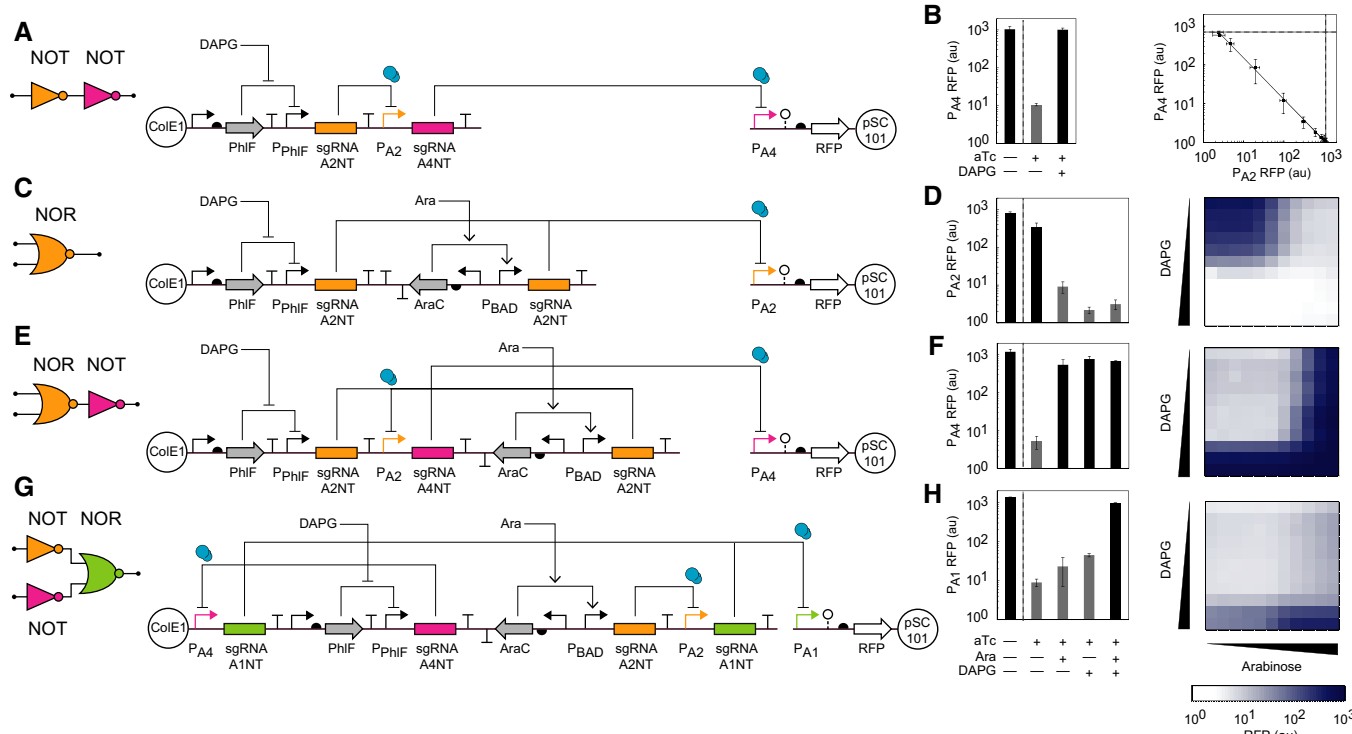

**Figure 3.   Design and characterization of synthetic circuits.**

A   The wiring diagram and genetic schematic for a double inverter circuit are shown. The sgRNA-A2NT/P$_{A2}$ pair is shown in orange, the sgRNA-A4NT/P$_{A4}$ pair is shown in magenta, dCas9 is shown in blue, positive regulation is indicated by arrows, and negative regulation is indicated by flat-headed arrows.

B   The digital RFP response of the NOT-NOT gate is shown for the two input inducer states (dCas9 induced with 0.625 ng/ml aTc): no DAPG and 25 μM DAPG. Also shown is the RFP output without dCas9 induction (leftmost column), which represents the maximum achievable output. Gray columns are expected to be OFF, and black columns are expected to be ON (left). The trade-off in expression between the middle and output promoters (P$_{A2}$ and P$_{A4}$, respectively) is shown for intermediate sgRNA induction levels (right). DAPG concentrations from left to right (μM) are the following: 0, 2.42, 3.39, 4.74, 6.64, 9.30, 13.0, 18.2, 25.5, 35.7, and 50. Dashed lines are uninduced dCas9 control experiments and represent the maximum output for each promoter. Error bars represent the standard deviation of three independent experiments on different days.

C   The wiring diagram and genetic schematic for a NOR(A,B) gate are shown. The sgRNA-A2NT/P$_{A2}$ pair is shown in orange, and dCas9 is shown in blue.

D   The NOR gate digital RFP response is shown (left) for the four input inducer states (with dCas9 induced by 0.625 ng/ml aTc): no arabinose or DAPG, arabinose (2 mM), DAPG (25 μM), and arabinose and DAPG (2 mM and 25 μM). Also shown is the output without dCas9 induction (leftmost column). In addition, the circuit response to intermediate inducer values is shown to the right.

E   The wiring diagram and genetic schematic for a layered NOT[NOR(A,B)] gate (i.e. an OR gate) are shown. The sgRNA-A2NT/P$_{A2}$ pair is shown in orange, the sgRNA-A4NT/P$_{A4}$ pair is shown in magenta, and dCas9 is shown in blue.

F   The OR digital RFP response is shown (left) for five input inducer states (as in D). Intermediate values are also shown (right).

G   The wiring diagram and genetic schematic for a four sgRNA circuit with NOR[NOT(A),NOT(B)] functionality (i.e. an AND gate) are shown. The sgRNA-A2NT/P$_{A2}$ pair is shown in orange, the sgRNA-A4NT/P$_{A4}$ pair is shown in magenta, the sgRNA-A1NT/P$_{A1}$ pair is shown in green, and dCas9 is shown in blue.

H   The AND gate digital RFP response is shown (left) for five input inducer states (as in D). Intermediate values are also shown (right).

Data information: For graded induction of circuits in (D), (F), and (H), aTc was added to 0.625 ng/ml; arabinose was added to the following final concentrations (mM): 0, 0.00391, 0.00781, 0.0156, 0.0313, 0.0625, 0.125, 0.25, 0.5, 1, and 2; 2,4-diacetylphloroglucinol was added to the following final concentrations (μM): 0, 0.0244, 0.0488, 0.0977, 0.391, 0.781, 1.56, 3.13, 6.25, 12.5, and 25.

experiment, the trade-off between P$_{A2}$ and P$_{A4}$ expression can be seen at intermediate sgRNA induction levels (Fig 3B, right). The log-linear response curve spans almost three orders of magnitude.

In addition to layering, the construction of more complex circuits requires that gates be able to receive multiple inputs. So-called "Boolean-complete" logic gates—NOR and NAND functions—are particularly useful because they can be connected to build any computational operation. Genetic NOR gates have proven to be particularly easy to build using transcriptional regulation where two input promoters drive the expression of a repressor that turns off an output promoter. The capacity for the orthogonality of

sgRNA:promoter interactions has the potential to enable a very large number of NOR gates, which could be used to realize large integrated circuits. However, to date, it has not been shown that sgRNA-based gates can be designed to respond to more than one input promoter.

To build a simple NOR gate, we connected two input promoters to the transcription of independent copies of sgRNA-2NT (Fig 3C), either of which will repress a single output promoter (P$_{A2}$). These two input promoters are responsive to small molecule inducers: DAPG (P$_{PhlF}$) and arabinose (P$_{BAD}$). In the presence of dCas9, but neither arabinose nor DAPG, the NOR gate output from promoter

$P_{A2}$ remains high at only 2.3-fold reduction compared to the dCas9 (—) control due to leaky sgRNA production. When both inducers are added, there is 100-fold repression of the output promoter (Fig 3D), which is on par with the best gates that use protein-based repressors. The OFF state is ~threefold higher when only arabinose is added, which is likely due to the lower maximum activity from the $P_{BAD}$ promoter compared to $P_{PhlF}$. While this does not significantly degrade the function of the NOR gate alone, it is representative of the sensitivity of sgRNA-based gates to the dynamic range of the inputs and is potentially problematic when building longer cascades.

Next, we connected multiple NOR and NOT gates to build larger layered circuits. First, we built a simple circuit that inverts the output of the NOR gate to make an OR gate (Fig 3E). The $P_{A2}$ output of the NOR gate is used to drive the transcription of sgRNA-A4NT, which in turn represses the $P_{A4}$ output promoter. A challenge that emerged from building these circuits is transcriptional readthrough, which occurs because the output promoters are strong and the sgRNAs short. To mitigate this, strong unique terminators (Chen *et al*, 2013) are placed after each sgRNA, immediately downstream from the dCas9 handle region of the sgRNA (Qi *et al*, 2013). For the OR gate, the TrrnB and L3S2P55 terminators [terminator strengths, $T_S = 84$ for TrrnB and $T_S = 260$ for L3S2P55, respectively (Chen *et al*, 2013)] are placed after the two sgRNA-A2NT sequences, and L3S2P21 ($T_S = 380$) is placed after sgRNA-A4NT. The output of the OR gate is strongly repressed > 100-fold in the absence of both inducers compared to all other states (Fig 3F).

We then built a larger circuit by connecting three gates based on four sgRNAs. A cascade with two branches is formed by the A2NT and A4NT sgRNAs, which invert the output of the arabinose- and DAPG-inducible systems, respectively (Fig 3G). The output promoters from these NOT gates then connect to a NOR gate by using each to drive a different copy of sgRNA-A1NT. The computing portion of the circuit requires 1,234 nt to encode. This circuit should produce an AND logic operation, and, indeed, there is a 107-fold difference between the OFF and ON states when both inducers are absent and present (Fig 3H). There is some leakiness when either input is induced alone, and these states show 2.6- to 5.0-fold activity above the OFF state observed in the absence of both inducers. Four versions of this circuit were designed with varied sgRNA positions and orientations. Other versions were slightly less functional, with higher OFF states and lower ON state; the best version is presented here. This circuit can be compared to a similar AND gate design built from TetR homologues. That circuit generated a ~fivefold response and required 2,577 nt to encode (Stanton *et al*, 2014).

### Interfacing the synthetic circuit with a native *Escherichia coli* regulatory network

Guide RNAs can be designed to knock down genes encoded in the host genome (Qi *et al*, 2013). In this way, native cellular processes can be easily actuated as an output of an sgRNA-based circuit using the same biochemistry. To demonstrate this, we started with the OR circuit (Fig 3E) and substituted the sgRNA used for the NOT gate with one designed to target the *malT* gene in the *E. coli* genome (Fig 4A). MalT is a positive regulator of the maltose utilization operons. A knockdown would alter sugar utilization and has additional impacts on the cellular phenotype (Tchetina & Newman, 1995; Boos & Bühm, 2000). Notably, it decreases the production of

LamB—the lambda phage receptor—resulting in decreased susceptibility of *E. coli* to lambda phage infection (Thirion & Hofnung, 1972). To target *malT*, we designed sgRNA-MalT-3NT to target the non-template strand of the protein coding sequence from the 110th to the 117th codon. By targeting the non-template strand, the roadblock formed by dCas9 would disrupt any transcription from upstream promoters (Bikard *et al*, 2013; Qi *et al*, 2013).

Cells harboring this circuit exhibit a 240-fold reduction in lambda plaque formation in the absence of both inducers (Fig 4C). When either or both inducers are present, the cells show near-wild-type phage infectivity. In addition, we can separately report the activity of an internal state of the circuit by using $P_{A2}$, which is the output of the NOR gate alone, to drive the transcription of RFP. This results in a NOR gate that is repressed 120-fold when either inducer is present (Fig 4B). These experiments demonstrate that a heterologous output (knockdown of RFP) and an endogenous response (knockdown of MalT) can be simultaneously co-regulated according to different logic operations using the same underlying circuit.

## Discussion

Extensible NOR and NOT gates are fundamental logic operations from which more complex circuitry can be built. Previously, these gates have been based on transcription factors that bind to DNA, such as phage repressors, LacI, and TetR homologues. Gates based on dCas9 and guide RNAs offer several advantages. The most significant is the ease by which new sgRNA:promoter pairs can be designed and the orthogonality that they exhibit with each other. While there has been much discussion regarding off-target Cas9 interactions and several efforts seeking to reduce it (Cradick *et al*, 2013; Fu *et al*, 2013, 2014; Hsu *et al*, 2013; Mali *et al*, 2013; Pattanayak *et al*, 2013; Ran *et al*, 2013; Guilinger *et al*, 2014; Kuscu *et al*, 2014; Tsai *et al*, 2014; Wu *et al*, 2014), this is not as relevant for synthetic circuits because sgRNAs can be designed to be maximally different from each other and the host genome. Indeed, no designed sgRNAs had to be discarded from the orthogonal set that we built, either for activity, orthogonality, or growth defects. Further, one transcriptomic analysis of CRISPR interference revealed no off-target signatures (Qi *et al*, 2013). This is a major improvement over the protein-based gates, which have problems in all of these areas. The "operator" that is bound by the sgRNA seed region is also relatively small (13 bp) and can be easily inserted between the -10 and -35 region of a promoter (TetR homologue operators range from 20 to 50 bp). In addition, the gates are small and can be easily synthesized as oligos, including in pooled libraries (Kosuri *et al*, 2013). The gates also reliably produce > 50-fold dynamic ranges. This is akin to the best protein-based gates, but those exhibit far more diversity in the leakiness, dynamic range, and shape of the response function.

Toxicity is observed from dCas9, where high levels reduce cell growth in *Escherichia coli*. While the mechanism of toxicity is still unclear, it has been reported to be more severe in other species. This may reduce the long-term evolutionary stability of dCas9 in engineered cells, as has been observed for other toxic genetic circuits (Sleight *et al*, 2010; Chen *et al*, 2013; Sleight & Sauro, 2013). However, we find that the toxicity can be managed by controlling the level of expression while still eliciting a substantial circuit response. Also, once dealt with, we do not observe substantial toxicity

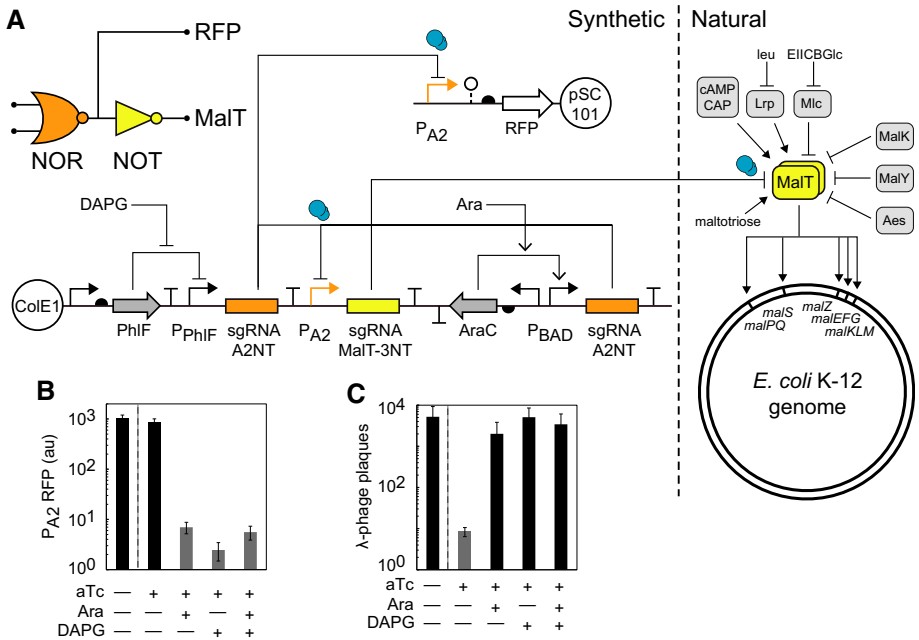

**Figure 4.  Interfacing logic circuits with host physiology.**

A   The wiring diagram and genetic schematic for a NOT[NOR(A,B)] gate are shown (i.e. an OR gate). The sgRNA-A2NT/P$_{A2}$ pair is shown in orange, the sgRNA-A4NT/P$_{A4}$ pair is shown in magenta, dCas9 is shown in blue, and both sgRNA-MalT-3NT and the *malT* gene are shown in yellow.

B   The NOR gate digital RFP response is shown for the four input inducer states (with dCas9 induced by 0.625 ng/ml aTc): no input inducer, arabinose (2 mM), DAPG (25 μM), and arabinose and DAPG (2 mM and 25 μM). Also shown is the output without dCas9 induction (leftmost column). Gray columns are expected to be OFF, and black columns are expected to be ON. Error bars represent the standard deviation of three independent experiments on different days.

C   The OR gate digital lambda phage infectivity response is shown for five input inducer states (as in B), where infectivity is measured by the number of lambda phage plaques formed on a bacterial lawn on an agar plate. Error bars represent the standard deviation of three independent experiments on different days.

as more sgRNAs are transcribed. This is in contrast to protein-based gates, which may have less toxicity individually, but can be problematic if multiple repressors are used in a design because their growth defects often stack and become severe.

There are also some challenges in working with dCas9 that are unique compared to protein-based gates. The shape of the response function, where no saturation is observed at high or low levels, poses a problem when layering gates. Without nonlinearity, the signal is degraded at each layer. Indeed, we attempted to add another layer to the AND gate, and this yielded a non-responsive circuit likely for this reason. Because there is no RBS to tune, it is difficult to fix this problem through the rational modification of the gate. No cooperativity also impedes the use of these gates for dynamic and multistable circuits, such as bistable toggle switches, pulse generators, or oscillators. Adding cooperativity could potentially be accomplished through dCas9 dimerization to effect promoter looping, sgRNA feedback latching motifs, or sequestration-based techniques such as "decoy operators" to titrate sgRNA away from cognate promoters. While the graded response could be of value for analog circuit construction, an inability to change its shape could be problematic. It may be possible to change the position of the response function by engineering specific mismatches to reduce the effectiveness of repression (Farzadfard *et al*, 2013). In addition, it is more difficult to connect input promoters upstream in series before an sgRNA, which has been a valuable design strategy for protein-based gates. Doing this would both require processing to remove the 5′-mismatch from the sgRNA, and also minimization of

transcriptional roadblocking, which could occur at the downstream promoter. Finally, because all of the gates require the same dCas9, this could impose retroactivity in the system where the activity state of upstream gates impacts the performance of downstream gates. An approach to circumvent this for larger circuits may be to use multiple orthogonal Cas9 homologues in a design (Esvelt *et al*, 2013).

It has been challenging to build genetic circuits that are as robust or capable as their natural counterparts. The potential for dCas9 to address this problem is vast. Synthetic sgRNAs can be designed to target a large number of sequences—synthetic and natural—and the sgRNA circuit architecture can be encoded in compact genetic constructs. This could allow the paradigm of analog and digital computing to be applied *in vivo* without requiring large and cumbersome constructs. dCas9 circuits also offer a mechanism whereby the same biochemistry can be used both to build circuitry that is orthogonal to the host and to directly interface host processes by design.

# Materials and Methods

### Strains and media

*Escherichia coli* DH10b (F– *mcr*A Δ(*mrr-hsd*RMS-*mcr*BC) Φ80*lac*ZΔM15 Δ*lac*X74 *rec*A1 *end*A1 *ara*D139 Δ(*ara leu*) 7697*gal*U *gal*K*rps*L*nup*G λ–) (Durfee *et al*, 2008) was used for cloning (New

England Biolabs, MA, C3019). *Escherichia coli* K-12 MG1655* [F-λ-ilvG- rfb-50 rph-1 Δ(araCBAD) Δ(LacI)] (Blattner *et al*, 1997) was used for measurement experiments. Cells were grown in LB Miller broth (Difco, MI, 90003-350) for overnight growth and cloning, and MOPS EZ Rich Defined Medium (Teknova, CA, M2105) with 0.4% glycerol carbon source for measurement experiments. Ampicillin (100 μg/ml), kanamycin (50 μg/ml), and spectinomycin sulfate (50 μg/ml) were used to maintain plasmids. Arabinose (Sigma Aldrich, MO, A3256), 2,4-diacetylphloroglucinol (Santa Cruz Biotechnology, TX, CAS 2161-86-6), and anhydrotetracycline (aTc) (Sigma Aldrich, MO, 37919) were used as chemical inducers. The fluorescent protein reporters YFP (Cormack *et al*, 1996) and mRFP1 (Campbell *et al*, 2002) were measured with cytometry to determine gene expression.

### Flow cytometry analysis

Fluorescent protein production was measured using the LSRII Fortessa flow cytometer (BD Biosciences, San Jose, CA). Between $10^4$ and $10^5$ events were collected for subsequent analysis with the software tool FlowJo v10 (TreeStar, Inc., Ashland, OR). From the resulting fluorescence histograms for YFP and RFP, we calculated the geometric means of each sample and then corrected for cellular autofluorescence by subtracting the geometric mean of a strain harboring only pAN-$P_{Tet}$-dCas9 that was grown in an identical manner.

### Computational design of sgRNA-promoter pairs

DNA sequences of 13 nucleotides in length were generated using the Random DNA Sequence Generator (http://www.faculty.ucr.edu/~mmaduro/random.htm), with a GC content probability parameter of 0.5. The resulting sequences were flanked by forward and reverse PAMs and the −35 and −10 sigma factor binding sites to generate sgRNA-repressible promoters. If the forward sequence for the promoter contained any stretches with more than three guanine nucleotides, the promoter design was discarded due to the difficulty in synthesizing oligos with G-quadruplexes (Burge *et al*, 2006). Next, the 12 nucleotides adjacent to either the forward or reverse PAM were searched for in the genome of *E. coli* strain K-12 substrain MG1655 (taxid: 511145) using Standard Nucleotide BLAST (http://blast.st-va.ncbi.nlm.nih.gov/Blast.cgi?PROGRAM=blastn) (Altschul *et al*, 1990) to search for somewhat similar sequences (blastn). The following parameters were used: Short queries were enabled; expect threshold = 10; word size = 11; match/mismatch scores = 2–3; gap costs = existence: 5, extension: 2; and low complexity regions unmasked. Of the ten sgRNAs designed, no 12 nt seed regions had complete homology to a PAM-adjacent locus in the *E. coli* genome. If the resulting 20 nucleotide sgRNAs had GC content < 35% or > 80%, the sequence was discarded and redesigned.

### Induction endpoint assays

*Escherichia coli* MG1655* cells were transformed with three plasmids encoding: (i) inducible dCas9, (ii) one or more sgRNAs, and (iii) a fluorescent reporter. Cells were plated on LB agar plates with appropriate antibiotics. Transformed colonies were inoculated into MOPS EZ Rich Defined Medium with 0.4% glycerol and appropriate antibiotics and were then grown overnight in V-bottom 96-well plates (Nunc, Roskilde, Denmark, 249952) in an ELMI Digital Thermos Microplates shaker incubator (Elmi Ltd, Riga, Latvia) at 1,000 rpm and 37°C. The next day, cultures were diluted 180-fold into EZ Rich Medium with antibiotics and grown with the same shaking incubator parameters for 3 h. At 3 h, cells were diluted 700-fold into EZ Rich Medium with antibiotics and inducers. The cells were grown using the same shaking incubator parameters for 6 h. For cytometry measurements, 40 μl of the cell culture was added to 160 μl of phosphate-buffered saline with 0.5 mg/ml kanamycin to arrest cell growth. The cells were placed in a 4°C refrigerator for 1 h to allow the fluorophores to mature prior to cytometry analysis.

### Toxicity measurements

For dCas9 toxicity measurements, cells were grown identically to the induction endpoint assays until the second dilution after the 3-h growth. From here, the cultures were diluted 360-fold into EZ Rich Defined Medium with 0.4% glycerol with antibiotics and inducers in 2 ml 96-deep well plates (USA Scientific, FL, 1896-2000) and were grown for 6 h in a Multitron Pro shaker incubator (*In Vitro* Technologies, VIC, Australia) at 37°C and 1,000 rpm. At this point, cultures were transferred to 1-cm optical cuvettes, and the cultures optical density at 600 nm was measured for the cell cultures, after a blank measurement with EZ Rich Medium. For sgRNA toxicity measurements, cells were grown identically to the induction endpoint assays.

### Induction timecourse assays

Timecourse experiments were performed identically to endpoint assays, with the exception that cells were grown in 14-ml round-bottom polystyrene culture tubes (VWR, PA, 60819-524). After the second dilution into inducers, culture samples were taken every 30 min for 7 h and were added to phosphate-buffered saline with 0.5 mg/ml kanamycin for subsequent cytometry analysis.

### Inducer concentrations

For dCas9 toxicity measurements, arabinose was added to 2 mM, and aTc was added to the following final concentrations (ng/ml): 0.0391, 0.313, 0.625, 1.25, 5, and 10. For sgRNA response curve experiments, aTc was added to 0.625 ng/ml and arabinose was added to the following final concentrations (mM): 0, 0.03125, 0.0625, 0.125, 0.25, and 0.5. For timecourse and orthogonality experiments, aTc was added to 0.625 ng/ml and arabinose was added to 2 mM. For digital genetic circuit measurements and lambda phage infection experiments, inducers were either absent or added to the following final concentrations: 0.625 ng/ml aTc, 2 mM arabinose, and 25 μM 2,4-diacetylphloroglucinol. For the intermediate genetic circuit measurements, aTc was added to 0.625 ng/ml; arabinose was added to the following final concentrations (mM): 0, 0.00391, 0.00781, 0.0156, 0.0313, 0.0625, 0.125, 0.25, 0.5, 1, and 2; 2,4-diacetylphloroglucinol was added to the following final concentrations (μM): 0, 0.0244, 0.0488, 0.0977, 0.391, 0.781, 1.56, 3.13, 6.25, 12.5, and 25.

**Lambda phage infection assay**

*Escherichia coli* MG1655* cells were grown from colonies overnight in EZ Rich Defined Media with antibiotics. The next day, cultures were diluted 180-fold into EZ Rich Medium with 0.4% glycerol and antibiotics and grown at 37°C shaking at 250 rpm in culture tubes for 3 h. Next, cells were diluted 180-fold once again into five different tubes of 4 ml of EZ Rich Medium with antibiotics and containing the five different inducer conditions. These cells were grown for 6 h using the same shaking incubator conditions in culture tubes. After 6 h, each culture was pelleted at 4,000 *g* and then resuspended in 100 µl of 10 mM $MgSO_4$. Half of each resuspension (50 µl) was diluted into 950 µl of 10 mM $MgSO_4$, and the optical density at 600 nm was measured. The remaining 50 µl of each cell resuspension was diluted to an $OD_{600}$ of 3.0 in 10 mM $MgSO_4$. Next, 1 µl of lambda phage was added to 100 µl of each cell resuspension, vortexed lightly, and then allowed to incubate at 37°C for 1 h. Finally, all 100 µl of cells were plated onto 1.5% agar LB Miller plate and allowed to grow overnight at 37°C. The next day, phage plaques were counted on each plate.

**Supplementary information** for this article is available online: http://msb.embopress.org

## Acknowledgements

CAV is supported by the Defense Advanced Research Projects Agency (DARPA CLIO N66001-12-C-4016), US National Institutes of Health (GM095765), the US National Institute of General Medical Sciences (NIGMS grant P50 GMO98792), and US National Science Foundation (NSF) Synthetic Biology Engineering Research Center (SynBERC EEC0540879). CAV and AAKN are supported by the Defense Advanced Research Project Agency (DARPA CLIO N66001-12-C-4018) through Ginkgo BioWorks and the Office of Naval Research (ONR) Multidisciplinary University Research Initiative (MURI Grant N00014-13-1-0074; Boston University MURI award 4500000552). AAKN receives Government FA9550-11-C-0028 and is awarded by the Department of Defense, Air Force Office of Scientific Research, National Defense Science and Engineering Graduate (NDSEG) Fellowship, 32 CFR 168a.

## Author contributions

CAV and AAKN conceived of the study and designed the experiments. AAKN performed the experiments and analyzed the data. CAV and AAKN wrote the manuscript. CAV managed the project.

## Conflict of interest

The authors declare that they have no conflict of interest.

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
