## [Review Process File · Molecular Systems Biology]

Multi-input CRISPR/Cas genetic circuits that interface host regulatory networks

Alec Nielsen

Corresponding author: Christopher Voigt, Massachusetts Institute of Technology

Review timeline:

Submission date:	01 September 2014
Editorial Decision:	26 September 2014
Revision received:	26 October 2014
Accepted:	29 October 2014

Editor: Maria Polychronidou

Transaction Report:

1st Editorial Decision

26 September 2014

Thank you again for submitting your work to Molecular Systems Biology. We have now heard back from the three referees who agreed to evaluate your manuscript. As you will see from the reports below, the reviewers acknowledge that the presented strategy and circuits are potentially useful. However, they raise a series of concerns, which should be carefully addressed in a revision of the manuscript.

One of the more substantial issues was raised by reviewer #1, and refers to the need to improve and broaden the applicability of the presented tools. This reviewer provides constructive suggestions regarding this point. Without repeating all of the reviewers' comments, most of them refer to the need to include text modifications and to further explain and/or discuss several points throughout the manuscript. Importantly, both reviewers #1 and #2 indicate that a comparison of the presented circuit with existing repressors should be included. Moreover, reviewer #3 recommends avoiding the use of jargon, in order to make the main findings easily accessible to a broader audience.

If you feel you can satisfactorily deal with these points and those listed by the referees, you may wish to submit a revised version of your manuscript. Please attach a covering letter giving details of the way in which you have handled each of the points raised by the referees. A revised manuscript will be once again subject to review and you probably understand that we can give you no guarantee at this stage that the eventual outcome will be favorable.

Reviewer #1:

Summary:

Nielsen and Voigt present their findings on the potential for logic gates based on transcriptional

repression by nuclease-null Cas9 to serve as components for complex genetic circuits. Working in *E. coli*, they determine the optimal level of dCas9 expression that does not result in substantial toxicity, characterize the performance of NOT gates based on a small set of orthogonal sgRNAs, and combine them into NOR and AND gates.

Their AND gate is notably superior to previous attempts using TetR homologues. The authors also connect gates to malT transcription, demonstrating loss of lambda infectivity absent induction, while simultaneously measuring the activity of just one internal gate using a fluorescent reporter.

As the authors acknowledge, many of the strategies described here were first pioneered in mammalian cells. While bacteria may have more alternative tools available, it's true that the field of circuit engineering has always suffered from a dearth of truly orthogonal tools. The authors' advance using Cas9 offers a potential technical solution to at least one aspect of the problem. While they only build and test five different orthogonally repressible promoters, the authors extensively characterize their behavior, including some nice results on the lack of saturation and the associated issues posed by that finding for layering gates.

Major points:

The formidable challenge facing all genetic circuit characterization papers involves convincing other laboratories that the findings will be useful. Many labs can find a use for better inducible promoters (witness how many are still using vectors by Lutz and Bujard 1997), while rather fewer have applications requiring multiple logic gates. However, a highly effective inverter is another story. The 440-fold repression observed with one of the sgRNAs, plus the advantage of ready connection to existing cellular pathways, might be the most readily applicable finding of the study to other labs. I highly recommend emphasizing this and explicitly comparing it to other available NOT gates. No additional experiments would be necessary; a simple table listing relative fold repression should suffice.

The titratable promoters used in this work represent an additional barrier to adoption. It seems unlikely that even other circuit researchers will want to titrate aTc to 0.625 ng/mL in all their experiments, much less other laboratories; in addition, many will be using the p15a origin for other purposes. I suggest the authors 1) make a constitutively expressed dCas9 version on a plasmid, ideally one with a less common origin such as CDF or better yet incW, 2) make a constitutively expressed version integrated into the genome, and 3) tune the promoter and RBS strengths of each to be equivalent to their inducible plasmid at 0.625 ng/mL. This comparatively straightforward experiment (make a dozen variants for each, integrate the genomic ones by recombineering, Cre/lox, or lambda integrase, and assay using the YFP reporter) should not take more than a few weeks and would greatly increase the likelihood of other laboratories adopting the approach by making Cas9-based regulation compatible with any *E. coli* strain.

With respect to elaborations and possible improvements on the basic approach, it's surprising that the authors do not mention various published ways of processing sgRNAs to remove extraneous 5' nucleotides (one example is the already-cited reference 24), which have been shown to allow multiple sgRNAs to be produced from the same promoter and should also allow two promoters to drive a common sgRNA (which they state does not work in the introduction). I'd like to see them test whether this approach works in their system, but feel that would be too much to ask for in this work given that their approach does not require it. Instead, I would recommend that they mention the possibility in the discussion.

The overall phrasing and readability of the article could use some work. There are frequent repetitions ("The first plasmid controls the expression of dCas9, which can exhibit toxicity when overexpressed" "dCas9 can exhibit toxicity when overexpressed"), awkward sentences ("A potentially scalable approach is to use dCas9, which employs small guide RNAs (sgRNAs) to bind to DNA sequences with high fidelity, where it can function as a repressor"; "The potential for dCas9 to address this problem is vast. They are orthogonal to host regulation and have a small footprint..."), and lack of definitions (dCas9 is only defined in a figure caption) that don't do justice to previous examples of the senior author's writing. A bit of attention to clarity could do wonders here.

Minor points:

The authors do not specify which Cas9 ortholog was used. Presumably it is from *S. pyogenes* given the NGG PAM, but it could also be from *S. mutans* or *S. agalactiae* given that same information.

With respect to keywords, zinc-fingers and TALEs are not relevant to this work. I would recommend replacing them with "genetic circuit" and "gene regulation" or similar.

Conclusion:

Perhaps more than any other approach save refactoring, Cas9-based regulatory tools have the potential to rescue the field of genetic circuits in the eyes of the broader molecular biology community. The question is whether it's possible to overcome the important problems such as signal loss that the authors identify here - and whether the demand for complex circuits will eventually manifest. By focusing on more immediately applicable tools such as inverters and making their findings readily usable by other laboratories interested in connecting signals to endogenous networks, this research could build a bridge to a brighter future for genetic circuits - and remain of substantial utility even if that future proves elusive for some time to come. Given these minor revisions, I recommend publication in MSB.

Reviewer #2:

The manuscript from Nielsen and Voigt presents the implementation of molecular gates in *e.Coli* using dCas9/sgRNAs as the core transcriptional control mechanism. The manuscript is well written and the experimental results are strong.

The methods are important for synthetic biologists and the circuits can be useful for a large community of researchers.

I read the paper with great interest but there are a couple of general issues that dampened my enthusiasm.

Major Points:

1. I was not able to identify a technical advance in the use of dCas9/sgRNAs. In terms of engineering the paper is great, yet is this sufficient for MSB? I would argue that the concept and experimental findings are not conceptually novel anymore (Lu and Weiss papers, and a couple of IGEM teams).
2. The authors argue that the biggest advantage of using dCas9/sgRNAs is the ability to scale synthetic circuits, at low cost. While I generally agree, I don't think that the experiments provide associated evidence nor push the size limits for genetic architectures in bacteria.
3. The manuscript does not have any quantitative analysis or simulations.
4. A direct comparison of the genetic circuits implemented with the dCas9/sgRNA versus other well-characterized transcriptional repressors would strengthen the results/claims.

Minor Points:

1. Phrase "almost arbitrary DNA sequences".
> Please be more specific (i.e. PAM and other constraints).
2. "A 12bp seed region of the complementary domain has incredible fidelity, where Cas9 can discern single base pair changes."
> I would tone down statements about the specificity of Cas9. The tolerance for mismatches are now documented and indeed people are trying to investigate its general pattern. Examples include:
<http://www.ncbi.nlm.nih.gov/pubmed/24837660>
<http://www.ncbi.nlm.nih.gov/pubmed/24752079>
<http://www.ncbi.nlm.nih.gov/pubmed/23939622>
3. There are a number of phrases where the authors discuss about their mechanism as RNA

repression. For example: "Here, we build a set of transcriptional gates based on sgRNA repression of a synthetic E. coli promoter" and "There are also some challenges in working with dCas9 that are unique compared to protein based gates".

> The authors have to be more careful here, the mechanism of repression is RNA-guided and protein (dCas9) based.

Reviewer #3:

Summary:

The authors have adapted the dCas9-CRISPR system, which represses transcription in bacteria (i.e. NOT(A) logic in Figure 2), to create multiplex logic (i.e. NOR(A,B) in Figure 3). The authors then demonstrated that they can create AND logic circuit (i.e. NOR(NOT A, NOTB) = AND(A,B) in Figure 3). They then show that these logic circuits can interface with and regulate biological targets in Figure 4. All of this work was done in E. coli.

General remarks:

The manuscript is novel and interesting. The experiments were well-executed and show how NOT and NOR dCas9-CRISPR modules can be stitched together to build complex logics. This is a major technical advance. The synthetic biology hype promoted by Knight/Endy in the 2000s may have finally arrived thanks to dCas9-CRISPR and this work from the Voigt lab. This article will be of interest to systems biologists, synthetic biologists, and the broad community of engineers, mathematicians, and biologists.

Major points:

(**) The authors should provide details regarding plasmid construction.

(**) The authors should better discuss the toxicity problem. Figure 2b justifies their target dCas9 levels as a balance between maximum repression and minimal toxicity. However, even this target level of dCas9 has a 15% growth defect. This strikes me as a big problem, as mutations in dCas9 would quickly take over the population and disrupt the synthetic circuits. I don't expect the authors to fix this problem for the synthetic biology field, but they can improve upon their upbeat assessment that "toxicity can be managed by controlling the level of expression while still eliciting a substantial circuit response".

Please discuss the implications of dCas9 toxicity for evolutionary stability of the circuit. How might the field of synthetic biology mitigate this problem in the short- and long-term? Rational or directed evolution of dCas9?

(**) The authors state that sgRNA expression does not increase toxicity, but I would like to see the data.

(**) Bacterial genes induce quickly. Thus, I was surprised by the 1.5 hour repression lag (3 doubling times!) in Fig. 2f for a simple NOT gate. I don't expect the authors to solve the problem, but I would like a frank discussion as to whether they find this lag surprising (given previously published work). How much is due to RFP maturation? How much is due to slow dCas9/sgRNA assembly?

This poses a problem for scaling. More complex logics will require cascade of NOT and NOR gates. If each level requires 10 hours to reach maximum repression, then full computation could take several days or 100s of generations! Please comment on this serious problem.

(**) Please label axes of Fig. 3d,f,h heatmaps. The authors qualitatively indicate increasing and decreasing concentration of DPAG and L-arabinose, but we need to see the quantitative values. Are the x-y axes linear? Logarithmic (like Fig. 3b)? Are concentrations evenly spaced in linear or logarithmic coordinates?

Minor points:

- Drop the WIRED magazine jargon ... "disruptive technology", "orthogonal", etc..
- REFS for toxicity problem with synthetic circuits, on pg. 3?
- Unclear whether "Two variants of each sgRNA were built that target the non-template (-T) and template (-NT) strands" is a typo? Please check.
- What does "rail" mean on pg. 6?
- Figure 1 could better establish the connection between digital electronics and dCas9-based components. The authors repeatedly use engineering notation (triangles with dot = NOT, crescent-shield with dot = NOR, input-output lines as wires), which may not be familiar to the general readership.

Figure 1 could have 3 parts, each part educating the reader to the importance of upcoming Figures 2 (NOT), 3 (NOR), and 4 (NOTNOR = AND with real world application). Each part establishes a connection between digital electronic notation and dCas9 circuits, while presaging the upcoming Figures 2-4.

More specifically, remove (b) "golden gate cloning" to make space. Keep (a) as is, but now include an engineer triangle with dot = NOT above with "NOT" label above it. The new (b) would be explicit dCas9 circuit for NOR (i.e. Fig3c) with crescent shield above with "NOR" label. Part (c) is OK as is, but the authors can remove distracting circuit diagrams inside triangles / crescent because these details will be in new (a-b). The goal of (c) should be two part: (i) explain how combining NOT, NOR builds more complex logics (e.g. NOTNOR = AND) and (ii) dCas9 logic modules can be easily interfaced with host genome.

- Please make sure that all your color images are black & white printer friendly. For example, the heat map in Fig. 2d is unreadable in B&W.
- Do the +/- combinations in bar graphs of Fig. 3b,d,f,h correspond to the extreme corners of 2D heat maps in Fig. 3b,d,f,h? It would be helpful to indicate +/- combinations on the 2D heat maps.
- It would be helpful to write "NOT NOT(A) = A", "NOR(A,B)", "NOR(A,B) NOT = OR(A,B)", and "NOR(NOT A, NOTB) = AND(A,B)" for the general readership in Fig. 3a,c,e,g.
- Unnecessary information on Figure 4 (all the MalT regulators).

1st Revision - authors' response

26 October 2014

Reviewer #1:

1. [...] The 440-fold repression observed with one of the sgRNAs, plus the advantage of ready connection to existing cellular pathways, might be the most readily applicable finding of the study to other labs. I highly recommend emphasizing this and explicitly comparing it to other available NOT gates. No additional experiments would be necessary; a simple table listing relative fold repression should suffice.

We have updated the text to emphasize the large dynamic range of the sgRNAs and their ability to interface native cellular pathways. We have also listed each sgRNA's fold-repression values in the Supplemental Information. We have added a comparison between the response curves of sgRNAs and the TetR family of transcription factors to the Supplemental Information.

2. The titratable promoters used in this work represent an additional barrier to adoption. It seems unlikely that even other circuit researchers will want to titrate aTc to 0.625 ng/mL in all their experiments, much less other laboratories; in addition, many will be using the p15a origin for other

purposes. I suggest the authors 1) make a constitutively expressed dCas9 version on a plasmid, ideally one with a less common origin such as CDF or better yet incW, 2) make a constitutively expressed version integrated into the genome, and 3) tune the promoter and RBS strengths of each to be equivalent to their inducible plasmid at 0.625 ng/mL. This comparatively straightforward experiment (make a dozen variants for each, integrate the genomic ones by recombineering, Cre/lox, or lambda integrase, and assay using the YFP reporter) should not take more than a few weeks and would greatly increase the likelihood of other laboratories adopting the approach by making Cas9-based regulation compatible with any E. coli strain.

The suggested experiments are a significant effort that is outside the scope of this work and, while they may be helpful to those in the field, they would not impact the claims or conclusions of this manuscript. As such, we will not pursue these experiments as part of this work.

3. With respect to elaborations and possible improvements on the basic approach, it's surprising that the authors do not mention various published ways of processing sgRNAs to remove extraneous 5' nucleotides (one example is the already-cited reference 24), which have been shown to allow multiple sgRNAs to be produced from the same promoter and should also allow two promoters to drive a common sgRNA (which they state does not work in the introduction). I'd like to see them test whether this approach works in their system, but feel that would be too much to ask for in this work given that their approach does not require it. Instead, I would recommend that they mention the possibility in the discussion.

We have made the suggested revision to the fourth paragraph of the introduction: "Without additional RNA processing, this design does not work for sgRNA-circuits because of the detrimental influence of 5'-mismatches¹⁹ and the 'roadblocking' effect of CRISPRi (small for template-targeting sgRNAs, substantial for non-template-targeting sgRNAs)⁷. Hammerhead ribozymes and endoRNase cleavage of 5'-mismatches have both been shown to effectively remove extraneous 5'-RNA from sgRNAs^{23,24} and could be employed in multi-input dCas9 circuits".

3. The overall phrasing and readability of the article could use some work. There are frequent repetitions ("The first plasmid controls the expression of dCas9, which can exhibit toxicity when overexpressed" "dCas9 can exhibit toxicity when overexpressed"), awkward sentences ("A potentially scalable approach is to use dCas9, which employs small guide RNAs (sgRNAs) to bind to DNA sequences with high fidelity, where it can function as a repressor"; "The potential for dCas9 to address this problem is vast. They are orthogonal to host regulation and have a small footprint..."), and lack of definitions (dCas9 is only defined in a figure caption) [...].

We have edited the paper to improve the presentation, including the suggested edits.

4. The authors do not specify which Cas9 ortholog was used. Presumably it is from *S. pyogenes* given the NGG PAM, but it could also be from *S. mutans* or *S. agalactiae* given that same information.

We have clarified in the text that the dCas9 used is from Streptococcus pyogenes.

5. With respect to keywords, zinc-fingers and TALEs are not relevant to this work. I would recommend replacing them with "genetic circuit" and "gene regulation" or similar.

Keywords are helpful when researchers are searching for papers. We like to select words or phrases that are not otherwise prevalent in the text, but where the searcher might find cross interest. In this case, if they are searching for zinc finger proteins or TALEs and genetic circuits, then this work would be interesting and that is why these words were selected.

Reviewer #2:

1. A direct comparison of the genetic circuits implemented with the dCas9/sgRNA versus other well-characterized transcriptional repressors would strengthen the results/claims.

We have added a section in the Supplemental Information that compares the shape and dynamic

ranges of sgRNA repression to the family of TetR transcriptional repressors.

2. Phrase "almost arbitrary DNA sequences". Please be more specific (i.e. PAM and other constraints).

*We have edited this sentence to, "Genome editing has been revolutionized by the *Streptococcus pyogenes* Cas9 RNA-guided endonuclease due to its ability to target DNA sequences adjacent to 'NGG' motifs using a supplied guide RNA¹⁻⁶."*

3. "A 12bp seed region of the complementary domain has incredible fidelity, where Cas9 can discern single base pair changes." I would tone down statements about the specificity of Cas9. The tolerance for mismatches are now documented and indeed people are trying to investigate its general pattern. Examples include: <http://www.ncbi.nlm.nih.gov/pubmed/24837660>, <http://www.ncbi.nlm.nih.gov/pubmed/24752079>, <http://www.ncbi.nlm.nih.gov/pubmed/23939622>.

We have removed the sentence about Cas9 discerning single base pair changes in the sgRNA seed region. We have also added the suggested references to the section about off-target binding in the discussion.

3. There are a number of phrases where the authors discuss about their mechanism as RNA repression. For example: "Here, we build a set of transcriptional gates based on sgRNA repression of a synthetic *E. coli* promoter" and "There are also some challenges in working with dCas9 that are unique compared to protein based gates". The authors have to be more careful here, the mechanism of repression is RNA-guided and protein (dCas9) based.

We have made the suggested changes.

Reviewer #3:

1. The authors should provide details regarding plasmid construction.

We have expanded the section on plasmid construction in the 3rd paragraph of the introduction to read, "The entire transcription unit (promoter, sgRNA, and terminator) can be constructed from a pair of ≤ 200 nt single-stranded DNA oligonucleotides that anneal and extend each other at the dCas9 handle region. These ssDNA oligos also encode Type II restriction enzyme recognition sites that flank the transcription unit. The resulting dsDNA modules can then be combined into a final circuit plasmid using a one-pot Golden Gate assembly reaction²² (Figure 1b)."

2. The authors should better discuss the toxicity problem. Figure 2b justifies their target dCas9 levels as a balance between maximum repression and minimal toxicity. However, even this target level of dCas9 has a 15% growth defect. This strikes me a big problem, as mutations in dCas9 would quickly take over the population and disrupt the synthetic circuits. I don't expect the authors to fix this problem for the synthetic biology field, but they can improve upon their upbeat assessment that "toxicity can be managed by controlling the level of expression while still eliciting a substantial circuit response". Please discuss the implications of dCas9 toxicity for evolutionary stability of the circuit. How might the field of synthetic biology mitigate this problem in the short- and long-term? Rational or directed evolution of dCas9?

It is difficult to comment on the origins (or potential elimination) of the toxicity because the mechanism is unclear. We have added citations to publications that have explicitly investigated the evolutionary stability of genetic circuits.

3. The authors state that sgRNA expression does not increase toxicity, but I would like to see the data.

At the request of the reviewer, we have measured the toxicity due to sgRNA expression. This data is provided in the Supplemental Information and the main text has been revised.

4. Bacterial genes induce quickly. Thus, I was surprised by the 1.5 hour repression lag (3

doubling times!) in Fig. 2f for a simple NOT gate. I don't expect the authors to solve the problem, but I would like a frank discussion as to whether they find this lag surprising (given previously published work). How much is due to RFP maturation? How much is due to slow dCas9/sgRNA assembly? This poses a problem for scaling. More complex logics will require cascade of NOT and NOR gates. If each level requires 10 hours to reach maximum repression, then full computation could take several days or 100s of generations! Please comment on this serious problem.

This lag is not surprising and is consistent with other work. Each layer would require ~1 hour to compute and these layers could consist of many gates. The largest circuits to date consist of ~4 layers.

5. Please label axes of Fig. 3d,f,h heatmaps. The authors qualitatively indicate increasing and decreasing concentration of DAPG and L-arabinose, but we need to see the quantitative values. Are the x-y axes linear? Logarithmic (like Fig. 3b)? Are concentrations evenly spaced in linear or logarithmic coordinates?

We have added the quantitative numbers to the caption.

6. Drop the WIRED magazine jargon ... "disruptive technology", "orthogonal", etc..

We have edited the paper to reduce the jargon, but note that "orthogonal" is not jargon and necessary to put in the context of other work in the field.

7. REFS for toxicity problem with synthetic circuits, on pg. 3?

We have added references to this sentence.

8. Unclear whether "Two variants of each sgRNA were built that target the non-template (-T) and template (-NT) strands" is a typo? Please check.

We have corrected this sentence.

9. What does "rail" mean on pg. 6?

We have clarified this sentence to read, "This is exacerbated by the shape of the response functions for the sgRNA-based gates, which do not plateau at high or low input promoter levels."

10. Figure 1 could better establish the connection between digital electronics and dCas9-based components. The authors repeatedly use engineering notation (triangles with dot = NOT, crescent-shield with dot = NOR, input-output lines as wires), which may not be familiar to the general readership. Figure 1 could have 3 parts, each part educating the reader to the importance of upcoming Figures 2 (NOT), 3 (NOR), and 4 (NOTNOR = AND with real world application). Each part establishes a connection between digital electronic notation and dCas9 circuits, while presaging the upcoming Figures 2-4. More specifically, remove (b) "golden gate cloning" to make space. Keep (a) as is, but now include an engineer triangle with dot = NOT above with "NOT" label above it. The new (b) would be explicit dCas9 circuit for NOR (i.e. Fig3c) with crescent shield above with "NOR" label. Part (c) is OK as is, but the authors can remove distracting circuit diagrams inside triangles / crescent because these details will be in new (a-b). The goal of (c) should be two part: (i) explain how combining NOT, NOR builds more complex logics (e.g. NOTNOR = AND) and (ii) dCas9 logic modules can be easily interfaced with host genome.

We have clarified the gate nomenclature in the caption and text.

11. Please make sure that all your color images are black & white printer friendly. For example, the heat map in Fig. 2d is unreadable in B&W.

The heat map has been changed to be more legible upon B&W printing.

12. Do the +/- combinations in bar graphs of Fig. 3b,d,f,h correspond to the extreme corners of 2D heat maps in Fig. 3b,d,f,h? It would be helpful to indicate +/- combinations on the 2D heat maps.

The combinations of inducers used to make the heat maps and bar graphs have been added to the caption.

13. It would be helpful to write " $\text{NOT NOT}(A) = A$ ", " $\text{NOR}(A,B)$ ", " $\text{NOR}(A,B) \text{ NOT} = \text{OR}(A,B)$ ", and " $\text{NOR}(\text{NOT } A, \text{NOT } B) = \text{AND}(A,B)$ " for the general readership in Fig. 3a,c,e,g.

We have added this information in the caption.

14. Unnecessary information on Figure 4 (all the MalT regulators).

We like this figure because it clearly shows that MalT is a regulator embedded in a natural network and this is the linkage point with the synthetic circuitry.